# Understanding the Selective Extraction of the Uranyl Ion from Seawater with Amidoxime-Functionalized Materials: Uranyl Complexes of Pyrimidine-2-amidoxime [†]

**Sokratis T. Tsantis** [1,2,*] , **Zoi G. Lada** [1,2] , **Sotiris G. Skiadas** [1] , **Demetrios I. Tzimopoulos** [3] , **Catherine P. Raptopoulou** [4] , **Vassilis Psycharis** [4,*] and **Spyros P. Perlepes** [1,2,*]

1  Department of Chemistry, University of Patras, 26504 Patras, Greece; zoilada@iceht.forth.gr (Z.G.L.); up1055861@ac.upatras.gr (S.G.S.)
2  Institute of Chemical Engineering Sciences (ICE-HT), Foundation for Research and Technology-Hellas (FORTH), Platani, P.O. Box 144, 26504 Patras, Greece
3  Department of Chemistry, Aristotle University of Thessaloniki, 54124 Thessaloniki, Greece; dimtzim@auth.gr
4  Institute of Nanoscience and Nanotechnology, NCSR "Demokritos", 15310 Aghia Paraskevi Attikis, Greece; c.raptopoulou@inn.demokritos.gr
*  Correspondence: tsantis@iceht.forth.gr (S.T.T.); v.psycharis@inn.demokritos.gr (V.P.); perlepes@upatras.gr (S.P.P.)
†  Dedicated to Professor Albert Escuer for His Contributions to Inorganic Chemistry: A Great Scientist and a Precious Friend.

**Abstract:** The study of small synthetic models for the highly selective removal of uranyl ions from seawater with amidoxime-containing materials is a valuable means to enhance their recovery capacity, leading to better extractants. An important issue in such efforts is to design bifunctional ligands and study their reactions with *trans*-$\{UO_2\}^{2+}$ in order to model the reactivity of polymeric sorbents possessing both amidoximate and another adjacent donor site on the side chains of the polymers. In this work, we present our results concerning the reactions of uranyl and pyrimidine-2-amidoxime, a ligand possessing two pyridyl nitrogens near the amidoxime group. The 1:2:2 $\{UO_2\}^{2+}$/pmadH$_2$/external base (NaOMe, Et$_3$N) reaction system in MeOH/MeCN provided access to complex [UO$_2$(pmadH)$_2$(MeOH)$_2$] (**1**) in moderate yields. The structure of the complex was determined by single-crystal X-ray crystallography. The U$^{VI}$ atom is in a distorted hexagonal bipyramidal environment, with the two oxo groups occupying the *trans* positions, as expected. The equatorial plane consists of two terminal MeOH molecules at opposite positions and two N,O pairs of two deprotonated $\eta^2$ oximate groups from two 1.11000 (Harris notation) pmadH$^-$ ligands; the two pyridyl nitrogen atoms and the –NH$_2$ group remain uncoordinated. One pyridyl nitrogen of each ligand is the acceptor of one strong intramolecular H bond, with the donor being the coordinated MeOH oxygen atom. Non-classical C$_{aromatic}$-H···X (X=O, N) intermolecular H bonds and π–π stacking interactions stabilize the crystal structure. The complex was characterized by IR and Raman spectroscopies, and the data were interpreted in terms of the known structure of **1**. The solid-state structure of the complex is not retained in DMSO, as proven via $^1$H NMR and UV/Vis spectroscopic techniques as well as molar conductivity data, with the complex releasing neutral pmadH$_2$ molecules. The to-date known coordination chemistry of pmadH$_2$ is critically discussed. An attempt is also made to discuss the technological implications of this work.

**Keywords:** amidoxime-based ligands; coordination chemistry; spectroscopic (IR, Raman, UV/Vis, $^1$H NMR) studies; structural studies; uranium recovery from seawater; uranyl complexes

## 1. Introduction

There has been an explosive growth in the chemistry of the uranyl ion, *trans*-$\{U^{VI}O_2\}^{2+}$, during the last 15 years or so for various reasons, including aspects of molecular and metallosupramolecular chemistry [1]; the development of anionic uranyl complexes with large

pores for the removal of radiotoxic metal ions [2]; the chemical, thermal, and photochemical functionalization of the oxo group of this cation [3,4]; the use of uranyl complexes as selective homogenous catalysts [5]; the employment of such complexes in separation processes extremely useful in nuclear fuel processing and nuclear waste management [6]; and the unique photocatalytic activity of some uranyl compounds [7].

An interesting aspect of the uranyl ion is the chemistry associated with its recovery from seawater [8–11]. With the intense demand for nuclear power production, it is expected that the land-based deposits of U will have been depleted by the end of this century [8]. Surprisingly, the oceans contain huge quantities of this heavy metal ion. It is estimated that more than four billion metric tons of U are dissolved in the Earth's oceans, approximately 1000 times higher than all known terrestrial sources, a quantity enough to satisfy the nuclear power industry of the world for centuries. The metal in seawater is in the form of the complex anion $[UO_2(CO_3)_3]^{4-}$ associated with two $Ca^{2+}$ ions. A technological challenge is to discover insoluble high-performance adsorbents with functional groups, i.e., ligands that can remove $\{UO_2\}^{2+}$ ions at its seawater concentration of 3.3 ppb [8] in the presence of other competing ions, such as $Cu^{2+}$, $Ca^{2+}$, $Mg^{2+}$, and especially $\{V^{VI}O\}^{2+}$ and $\{V^{V}O_2\}^+$. From the materials tested, only polymeric poly(acrylamidoximes) are chemically stable and selective for uranyl uptake [8,12–16]. A general, simplified process is illustrated in Figure 1. The amidoxime derivatives are of three types depending on the reaction conditions (Figure 1a–c) [8,17]. These derivatives may have different binding abilities with the uranyl ion, which can affect the sorption ability and selectivity of the sorbent material. A successful strategy for uranium recovery from seawater has two steps: (i) the design of very active and highly selective uranyl chelating agents based on the amidoxime group and (ii) the incorporation of the ligands containing those molecular units to a polymeric matrix in a manner that significantly enhances the uranyl binding sites. The realization of this strategy requires, among others, molecular inorganic chemistry (coordination chemistry) approaches. Thus, few uranyl complexes with amidoxime-based ligands have been isolated and structurally characterized [13,16–27].

It is rather surprising that scientists have not well understood the reason for the strong and selective binding of the amidoxime group with $\{UO_2\}^{2+}$; this means that the exact nature of the binding mode is ambiguous, and various research groups have proposed different possibilities [8,27–31]. If the relationship between the amidoxime functionality on the polymeric sorbent materials and the details of their reactivity toward $\{UO_2\}^{2+}$ (and other competing metal ions) are revealed, then the U mining of the world's oceans might become a reality in the next 20–30 years!

We have recently embarked on a new program with the general goals of creating synthetic models for the highly selective removal of $\{UO_2\}^{2+}$ from seawater with amidoxime functionalized materials, designing bifunctional ligands containing amidoxime groups and a neighboring donor site, and performing their reactions with $\{UO_2\}^{2+}$; the second goal involves understanding the reactivity of polymeric sorbents possessing, in addition to the amidoxime group, a second different site on the side chains [32,33]. We describe here the reactions between uranyl sources and the ligand pyrimidine-2-amidoxime (IUPAC name: N′-hydroxypyrimidine-2-carboximidamide; another name is pyrimidine-2-carboxamide oxime), abbreviated as pmadH₂ (Figure 2). Our purpose is to investigate how the $\{UO_2\}^{2+}$ reactivity is affected by the number of the amidoxime groups and the positions of the heterocyclic nitrogen atoms in the aromatic ring; we hope that our studies will contribute to the understanding of the chemistry of polymeric sorbents possessing amidoxime group(s) and neighboring donor groups toward the selective removal of the uranyl ion from seawater. The coordination chemistry of pmadH₂ has been studied [34–44], but complexes of 5f metal ions have not been reported to date.

**Figure 1.** Preparation of polyacrylonitrile (PAN) side chains and their functionalization to form amidoxime derivatives: (**a**) open-chain diamidoxime moiety; (**b**) closed-ring glutarimide-dioxime moiety; (**c**) closed-ring glutarimidoxioxime moiety. Abbreviations: PE = polyethylene; RIGP = radiation-induced graft polymerization; PAN = polyacrylonitrile.

**pmadH$_2$**

**Figure 2.** The structural formula and abbreviation of the ligand used in this work are shown. The number of H atoms in the abbreviations denotes the potentially acidic hydrogens; it has been experimentally confirmed that, under certain conditions, one –NH$_2$ hydrogen can sometimes be removed under basic conditions upon coordination.

Compounds containing the amidoxime group -C(NH$_2$)=NOH have gained intense interest in recent years because of their involvement in supramolecular materials and

biological chemistry [45,46]; another interesting chemical aspect of such compounds is their tautomerism [47,48]. The coordination chemistry of amidoximes and their metal-involving reactions were reviewed a few years ago [49]. The amidoxime group can adopt several ligation modes as neutral or singly deprotonated at the oxime functionality; in all cases, the neutral $-NH_2$ group remains uncoordinated because it creates a mesomeric effect bearing a partial positive charge. This group can sometimes be acidic under basic conditions in the presence of metal ions; thus, the overall charge of the amidoxime can be $-2$, resulting in significant coordination flexibility [42].

Thus, the present work describes a complex prepared through the reaction of convenient uranyl sources and pmadH$_2$ with a variety of reactions. The product was characterized in the solid state using single-crystal X-ray crystallography, IR, and Raman spectroscopies. An attempt was also made to study the solution behavior of the compound in DMSO through molar conductivity measurements, UV/Vis spectroscopy, and $^1$H NMR spectroscopy.

## 2. Results and Discussion

### 2.1. Synthesis of the Complex

Seeking to contribute to the understanding of the chemistry associated with the reactivity of sorbents possessing an amidoxime group and a second donor site toward the uranyl, we performed the synthetic investigation of the $\{UO_2\}^{2+}$/pmadH$_2$ system.

Despite numerous efforts, we managed to isolate and structurally characterize only one product; this is $[UO_2(pmadH_2)(MeOH)_2]$ (**1**), which has a yellowish-orange color. The 1:1:2 reaction between $UO_2(NO_3)_2 \cdot 6H_2O$, pmadH$_2$, and NaOMe gave **1** in a moderate yield (~50%) (Equation (1)). The same complex (IR evidence) was obtained using Et$_3$N as base albeit in a lower yield (Equation (2)). The change in the reactants' molar ratio from 1:2:2 to 1:1:1 in the same solvent mixture again provided **1** in low yield. The binding mode and binding nature of a uranyl–amidoxime complex in an aqueous solution (to avoid the possible variations in the ligation mode caused by the crystallization process for the isolation of single crystals) was recently determined for the first time, using XAFS techniques combined with DFT calculations [28]. The results showed that, in a concentrated amidoxime solution, the local structure consists of a uranyl ion equatorially coordinated by three $\eta^2$ amidoximate groups. Stimulated by this surprising result, we attempted to prepare complexes containing the anion $[UO_2(pmadH)_3]^-$ by increasing the ratio of the ligand and the base. However, the 1:3:3 and 1:4:4 $UO_2(NO_3)_2 \cdot 6H_2O$:pmadH$_2$:Et$_3$N reaction mixtures yielded complex **1** again.

$$UO_2(NO_3)_2 \cdot 6H_2O + 2pmadH_2 + 2NaOMe \xrightarrow[MeCN]{MeOH} [UO_2(pmadH)_2(MeOH)_2] + 2NaNO_3 + 6H_2O \qquad (1)$$

$$UO_2(NO_3)_2 \cdot 6H_2O + 2pmadH_2 + 2Et_3N + 2MeOH \xrightarrow[MeCN]{MeOH} [UO_2(pmadH)_2(MeOH)_2] + 2(Et_3NH)(NO_3) + 6H_2O \quad (2)$$

The omission of the external base from the above reaction system gave a bordeaux powder. Its IR spectrum shows characteristic bands of the coordinated nitrates, and microanalytical data indicate the empirical formula $\{UO_2(NO_3)(pmadH)(pmadH_2)\}$. Efforts (employing a variety of crystallization techniques) failed to give single crystals of this product, and its study was not further pursued.

The reactions of $UO_2(O_2CMe)_2 \cdot 2H_2O$ and pmadH$_2$ are completely analogous. The 1:2:2 reaction between this uranyl source, the ligand, and NaOMe in MeOH gave single crystals of **1**, in a moderate yield (30–40%) (Equation (3) (IR evidence)). The reaction without NaOMe yielded an orange-red powder, whose microanalytical data are consistent with the formulation $\{UO_2(O_2CMe)(pmadH)(pmadH_2)\}_X$; in accordance with this empirical formula, the IR spectrum of the powder exhibits the $v_{as}(CO_2)$ and $v_s(CO_2)$ of the coordinated acetate groups at 1530 and 1395 cm$^{-1}$, respectively.

$$\text{UO}_2(\text{O}_2\text{CMe})_2 \cdot 2\text{H}_2\text{O} + 2\text{pmadH}_2 + 2\text{NaOMe} \xrightarrow[\text{MeCN}]{\text{MeOH}} [\text{UO}_2(\text{pmadH})_2(\text{MeOH})_2] + 2\text{NaO}_2\text{CMe} + 2\text{H}_2\text{O} \qquad (3)$$

### 2.2. Structure Determination of the Complex

The structure of **1** was determined by single-crystal X-ray crystallography. Crystallographic data are listed in Table 1. Selected bond distances and angles are given in Table 2, while hydrogen bonding details are summarized in Table 3. Structural plots are shown in Figures 3, 4, and S1.

**Table 1.** Crystallographic data and structural refinement parameters for complex **1**.

| Parameter | $[\text{UO}_2(\text{pmadH})_2(\text{MeOH})_2]$ (1) |
|---|---|
| Empirical formula | $C_{12}H_{18}UN_8O_6$ |
| Formula weight | 608.37 |
| Crystal system | triclinic |
| Space group | $P\bar{1}$ |
| Color | orange |
| Crystal size, mm | $0.18 \times 1.16 \times 0.11$ |
| Crystal habit | parallelepiped |
| $a$, Å | 7.4659(7) |
| $b$, Å | 7.9915(8) |
| $c$, Å | 8.7067(8) |
| $\alpha$, ° | 108.854(3) |
| $\beta$, ° | 103.664(3) |
| $\gamma$, ° | 104.709(3) |
| Volume, Å$^3$ | 445.82(7) |
| $Z$ | 1 |
| Temperature, K | 180 |
| Radiation, Å | Mo K$\alpha$, 0.71073 |
| Calculated density, g·cm$^{-3}$ | 2.266 |
| Absorption coefficient, mm$^{-1}$ | 9.15 |
| No. of measured, independent, and observed $[I > 2\sigma(I)]$ reflections | 13,557, 1946, 1946 |
| $R_{\text{int}}$ | 0.030 |
| Number of parameters | 152 |
| Final $R$ indices $[I > 2\sigma(I)]$ [a] | $R_1 = 0.0136, wR_2 = 0.0332$ |
| Goodness-of-fit on $F^2$ | 1.12 |
| Largest differences peak and hole (e Å$^{-3}$) | $1.07/-0.72$ |

[a] $R_1 = \Sigma(|F_o| - |F_c|)/\Sigma(|F_o|)$, $wR_2 = \{\Sigma[w(F_o{}^2 - F_c{}^2)^2]/\Sigma[w(F_o{}^2)^2]\}^{1/2}$, $w = 1/[\sigma^2(F_o{}^2) + (\alpha P)^2 + bP]$ where $P = [\max(F_o{}^2, 0) + 2F_c{}^2]/3$ ($\alpha = 0.0209$ and b = 0.1315).

**Table 2.** Selected bond lengths (Å) and angles (°) for complex $[\text{UO}_2(\text{pmadH})_2(\text{MeOH})_2]$ (**1**) [a].

| Bond Lengths (Å) | | Bond Angles (°) | |
|---|---|---|---|
| U1-O3 | 1.803(2) | O3-U1-O3′ | 180.0(1) |
| U1-O1 | 2.324(2) | O3-U1-O1 | 90.3(1) |
| U1-N1 | 2.405(2) | O3-U1-O1′ | 89.7(1) |
| U1-O2 | 2.455(2) | O3-U1-N1 | 90.4(1) |
| O1-N1 | 1.379(3) | O3-U1-O2 | 90.8(1) |
| N1-C1 | 1.293(3) | O1-U1-N1 | 33.9(1) |
| N2-C1 | 1.356(3) | O1-U1-O2 | 106.8(1) |
| C1-C2 | 1.484(3) | O2-U1-N1 | 73.0(1) |
| N3-C2 | 1.334(3) | O1-N1-C1 | 115.9(2) |
| N3-C5 | 1.344(3) | N1-C1-N2 | 122.7(2) |
| N4-C2 | 1.341(3) | N1-C1-C2 | 118.4(2) |
| N4-C3 | 1.339(3) | N2-C1-C2 | 118.8(2) |

[a] Primed atoms are generated using the symmetry operation $-x + 2, -y, -z$.

**Table 3.** Hydrogen-bonding interactions (Å, °) in the crystal structure of [UO$_2$(pmadH)$_2$(MeOH)$_2$] (**1**).

| D-H···A | d(D···A) | d(H···A) | <DHA | Symmetry Code of A |
|---|---|---|---|---|
| O2-H(O2)···N4 | 2.699(3) | 1.95(4) | 180(5) | |
| C3-H(C3)···O3 [a] | 3.446(3) | 2.59(4) | 158(3) | $x + 1, y, z$ |
| C5-H(C5)···N3 [a] | 3.497(3) | 2.69(3) | 152(3) | $-x + 3, -y - 1, -z + 1$ |

[a] Weak, non-classical hydrogen bonds. Abbreviations: D = donor; A = acceptor.

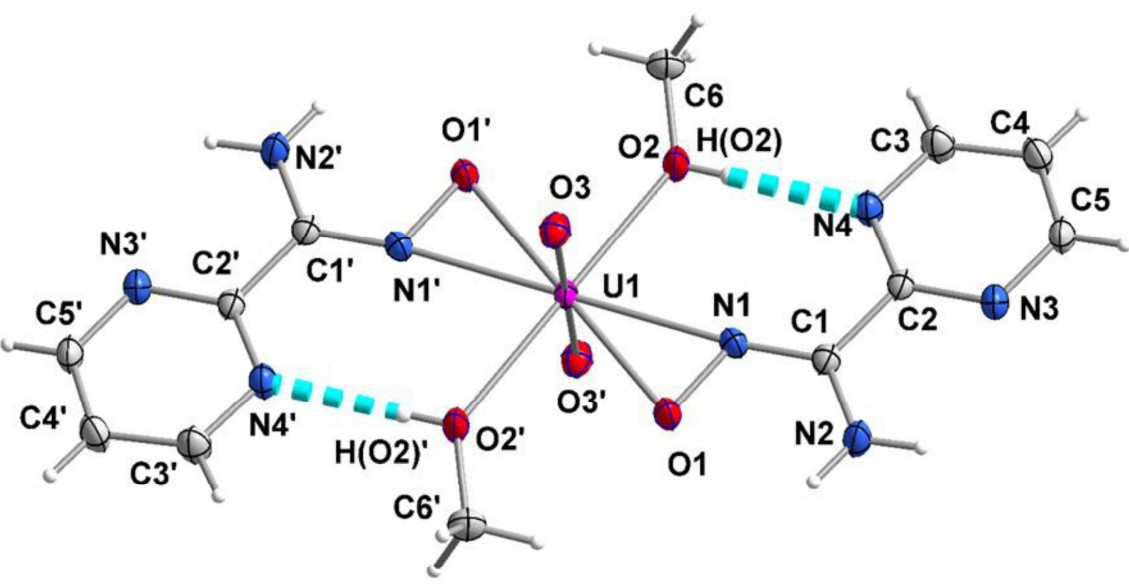

**Figure 3.** Partially labeled ORTEP-type plot of the molecular structure of compound [UO$_2$(pmadH)$_2$(MeOH)$_2$] (**1**). The dashed cyan lines represent the O2-H(O2)···N4 (and symmetry equivalent) hydrogen bond. Symmetry code: (′) $-x + 2, -y, -z$.

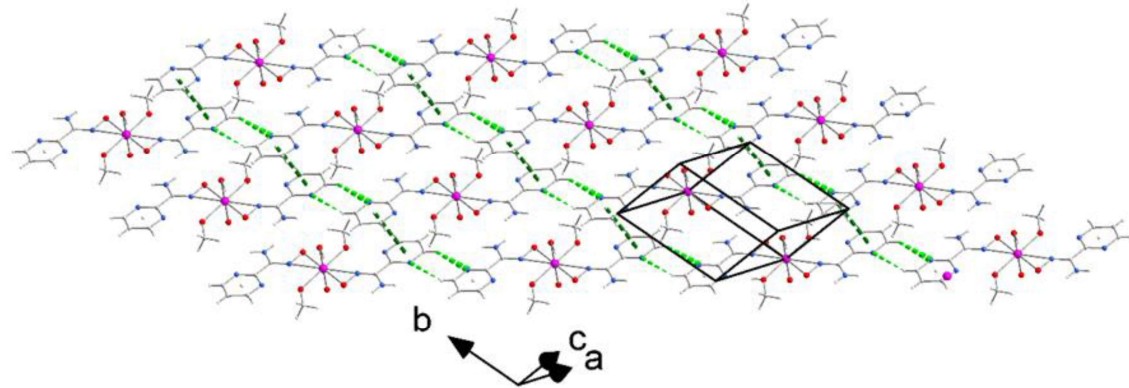

**Figure 4.** Layers of molecules parallel to the (10-1) crystallographic planes in the crystal structure of compound [UO$_2$(pmadH)$_2$(MeOH)$_2$] (**1**). Dashed light and dark green lines denote the C5-H(C5)···N3 (and symmetry equivalent) hydrogen bonds and π–π interactions, respectively. For metric parameters, see Table 3.

Compound **1** crystallizes in the centrosymmetric space group $P\bar{1}$. Its asymmetric unit contains a half-[UO$_2$(pmadH)$_2$(MeOH)$_2$] molecule since U1 is located on a crystallographically imposed inversion center; the molecule is thus centrosymmetric. U1 is surrounded by six oxygen and two nitrogen atoms in a very distorted hexagonal bipyramidal geometry. The two uranyl oxygen atoms (O3 and O3′) occupy the axial positions, as expected. The strictly planar (by symmetry) equatorial plane of the metal ion consists of two terminal methanol molecules (O2 and O2′) and two η$^2$ oximate groups (O1, N1. and their symmetry equivalents) from the singly deprotonated 1.11000 (Harris notation [50]) pmadH$^-$

ligands (Figure 5). The $-NH_2$ group and the two pyridyl nitrogen atoms of each ligand remain uncoordinated. Two of them (N4 and N4′) are acceptors of strong intramolecular H bonds, the donors being the coordinated MeOH oxygen atoms (O2 and O2′); see Figure 3 and Table 3. The distortion of the hexagonal bipyramidal coordination geometry at U1 is primarily a consequence of the small bite angles of the $\eta^2$ oximate groups (O1-U1-N1/O1′-U1-N1′ = ~34°).

**Table 4.** The to-date crystallographically characterized metal complexes of pmadH$_2$, pmadH$^-$, and pmad$^{2-}$, and the coordination modes of the amidoxime/amidoximate ligands [a].

| Complex [b,c] | Coordination Mode(s) | Ref. |
|---|---|---|
| {[Cd(pmadH$_2$)$_2$(L$^1$)]}$_n$ | 1.01100 | [43] |
| {[Cd(pmadH$_2$)(L$^2$)(H$_2$O)]}$_n$ | 1.01100 | [43] |
| {[Cd$_2$(L$^3$)(pmadH$_2$)$_2$(H$_2$O)$_2$]}$_n$ | 1.01100 | [39] |
| [MnCl$_2$(pmadH$_2$)$_2$] | 1.01100 | [41] |
| [MnCl(O$_2$CMe)(pmadH$_2$)$_2$] | 1.01100 | [40] |
| [Mn(O$_2$CMe)$_2$(pmadH$_2$)$_2$] | 1.01100 | [38] |
| {[Co(pmadH$_2$)$_2$(L$^1$)]}$_n$ | 1.01100 | [35] |
| {[Co(pmadH$_2$)$_2$(L$^4$)]}$_n$ | 1.01100 | [35] |
| {[Co(pmadH$_2$)(SO$_4$)(H$_2$O)$_2$]}$_n$ | 1.01100 | [35] |
| {[Mn(pmadH$_2$)$_2$(L$^4$)]}$_n$ | 1.01100 | [44] |
| {[Mn(pmadH$_2$)(L$^2$)(H$_2$O)]}$_n$ | 1.01100 | [44] |
| [Co$_3$(pmadH$_2$)$_6$](O$_2$CMe)$_2$(L$^4$)$_2$ | 2.1110 | [35] |
| [Co$_3$(pmadH$_2$)$_6$](O$_2$CMe)$_2$(L$^1$)$_2$ | 2.1110 | [35] |
| {[Cu(pmadH$_2$)(NO$_3$)](NO$_3$)}$_n$ | 1.01100 | [42] |
| {[Ni(pmadH$_2$)$_2$(L$^4$)]}$_n$ | 1.01100 | [34] |
| [Ni$_2$Mn(pmadH)$_6$](ClO$_4$) | 2.11100 | [37] |
| [Ni$_2$Fe(pmadH)$_6$](ClO$_4$) | 2.11100 | [37] |
| [Ni$_2$Tb(pmadH)$_6$](NO$_3$) | 2.11100 | [37] |
| [Ni$_4$(pmadH)$_4$(O$_2$CPh)$_2$(pyz)$_2$(MeOH)$_2$](ClO$_4$) | 3.21100 | [36] |
| [Cu$_4$(pmadH)$_2$(pmad)$_2$(NO$_3$)](NO$_3$) | 2.11100, 3.21110 | [42] |
| [Cu$_2$Ni$_2$(pmadH)$_2$(pmad)$_2$Cl$_2$] | 2.11100, 3.21110 | [42] |

[a] Using Harris notation; also see Figure 5. [b] Lattice solvent molecules have been omitted for clarity. [c] Abbreviations: L$^1$ = the dianion of 4-sulfonicbenzoic acid; L$^2$ = the dianion of 2-hydroxy-5-sulfonicbenzoic acid; L$^3$ = the tetranion of 1,1′:4′,1″-terphenyl-2′,4,4″,5′-tetracarboxylic acid; L$^4$ = the dianion of 3-sulfonicbenzoic acid; pyz = pyrazine.

**Figure 5.** The to-date crystallographically observed modes of pmadH$_2$ and its monoanionic (pmadH$^-$) and dianionic (pmad$^{2-}$) forms, and the Harris notation that describes these modes. The coordination modes are drawn with bold lines. The 1.11000 ligation mode was observed for the first time in complex **1** of this work. For more details, see Table 4.

The U=O bond length [1.803(2) Å] is slightly longer than the typical uranyl distances [18,21,26], due to the involvement of O3/O3′ in weak hydrogen-bonding interactions (*vide infra*). The U-O1/1′ bond is stronger than the U-O2/2′ (methanol) bonds. The O1-N1, N1-C1, and N2-C1 bond lengths indicate a partially delocalized amidoximate group (see the coordination mode 1.11000 in Figure 5).

At the supramolecular level, there are weak hydrogen bonds of the C(aromatic)-H···X (X=O, N) type and π–π stacking interactions. The molecules of **1** form chains parallel to the [1, −1, 1] directions through C5-H5···N3 hydrogen bonds. The chains develop layers parallel to the (10-1) crystallographic planes through the π–π stacking interactions

between centrosymmetrically related C2N3C5C4C3N4 and C2″N3″C5″C4″C3″N4″ rings [symmetry code: (″) $-x + 3, -y, -z + 1$] (Figure 4); the distance between the mean planes of neighboring rings is 3.37(1) Å. Neighboring layers interact through weak C3-H(C3)$\cdots$O3 (and symmetry equivalent), non-classical hydrogen bonds building the 3D architecture of the crystal structure (Figure S1).

Compound **1** is the first structurally characterized 5f element complex of pmadH$_2$ or its anionic forms. The previously characterized compounds and the coordination modes of the ligands are conveniently summarized in Table 4. The ligation modes with their Harris notations are illustrated in Figure 5. The novel coordination mode 1.11000 featuring a three-membered chelating ring was confirmed only in **1**. Almost all the previously reported (both homo- and heterometallic) complexes contain 3d metal ions. Exceptions are three polymeric Cd(II) compounds [39,43] and the mixed 3d/4f metal complex [Ni$_2$Tb(pmadH)$_6$](NO$_3$) [37]. Most of the complexes have the neutral ligand pmadH$_2$, which almost exclusively behaves as a N$_{pyridyl}$, N$_{oxime}$-bidentate chelating ligand (coordination mode 1.01100, Figure 5); two exceptions are the trinuclear Co(II) complexes [Co$_3$(pmadH$_2$)$_6$](O$_2$CMe)$_2$(L$^1$)$_2$·2H$_2$O and [Co$_3$(pmadH$_2$)$_6$](O$_2$CMe)$_2$(L$^4$)$_2$·0.5H$_2$O [35], in which the formally neutral ligand is in its zwitterionic form adopting the coordination mode 2.1110 (Figure 5) and bridging two metal ions. (L$^1$)$^{2-}$ and (L$^4$)$^{-2}$ are the dianions of the 4-sulfonic benzoic acid and 3-sulfonicbenzoic acid, respectively, acting as counterions. The monoanionic ligands pmadH$^-$ behave in the 2.11100 or 3.21100 manners (with the exception of the present complex **1**) [36,37,42]. Of particular interest are the tetranuclear clusters [Ni$_4$(pmadH)$_2$(pmad)$_2$(NO$_3$)](NO$_3$) and [Cu$_2$Ni$_2$(pmadH)$_2$(pmad)$_2$Cl$_2$] [42] which, in addition to two 2.11100 monoanionic (pmadH$^-$) ligands, also contain two novel dianionic (pmad$^{2-}$) groups that adopt the unusual 3.21110 mode involving the deprotonated amino moiety.

*2.3. Spectroscopic Characterization*

Complex **1** was characterized by a variety of spectroscopic and physical techniques in the solid state and in solution. Spectra are presented in Figures 6–8 and S2–S4. The IR spectrum shows two strong bands at 3456 and 3352 cm$^{-1}$, primarily assigned to the $v_{as}(NH_2)$ and $v_s(NH_2)$ modes, respectively [51]. The broad characteristic of the former band, accompanied by a weak shoulder at lower wavenumbers, might indicate that this band also involves the $v(OH)$ mode of coordinated MeOH. The $v_{as}(NH_2)$ and the $v_s(NH_2)$ bands appear at 3422 and 3328 cm$^{-1}$, respectively, in the IR spectrum of free pmadH$_2$. The small shifts of these bands compared to the corresponding bands of **1** are in agreement with the lack of involvement of -NH$_2$ in coordination. The bands at 1650 (medium intensity) and 942 (medium intensity) in the free pmadH$_2$ are due to the $v(C=N)$ and $v(N-O)$ vibrations, respectively [39,41,49]. These bands shift to lower [$v(C=N)$] and higher [$v(N-O)$] wavenumbers upon deprotonation and coordination. The IR spectrum of the complex shows the presence of a strong band at 870 cm$^{-1}$ assigned to the IR-active antisymmetric stretching vibration of the *trans*-{O=U=O}$^{2+}$ group ($v_3$) [51]. This band, which is absent from the spectrum of the free ligand, is red-shifted compared to the corresponding band for aquo uranyl complexes ($\sim$960 cm$^{-1}$), implying weaker U=O bonds in the complex. This probably arises [52] from the negative charge in the equatorial plane and the involvement of the oxo atoms (the so-called O$_{yl}$ atoms) in hydrogen-bonding interactions, which both weaken the axial uranium(VI)–oxygen bonds [52].

The $v_{as}(NH_2)$, $v(NH_2)$, and $v(OH)$ modes are hardly seen in the Raman spectra, as expected. The peaks at 1642 and 937 cm$^{-1}$ in the spectrum of pmadH$_2$ are assigned to the $v(C=N)$ and $v(N-O)$ vibrations, respectively [53,54]. The corresponding peaks in the spectrum of **1** appear at 1620 and 1002 cm$^{-1}$, the latter possibly overlapping with a pmadH$^-$ mode. Analogous coordination shifts are observed like those in the IR spectra. The strong peak at 820 cm$^{-1}$ is due to the Raman-active symmetric stretching vibration ($v_1$) of the uranyl ion [51]. The IR-active bending vibration $\delta$(O=U=O) [$v_1$] is expected to appear at $\sim$200 cm$^{-1}$, well below the lowest wavenumber limit of our spectrometer. The old and simplified empirical equation $0.933 \times v_3 = v_1$ [55], which has been derived for compounds

containing linear *trans*-$\{U^{VI}O_2\}^{2+}$ groups, applies very well for **1** (experimental $v_3$:870 cm$^{-1}$; experimental $v_1$:820 cm$^{-1}$; calculated $v_1$: 812 cm$^{-1}$).

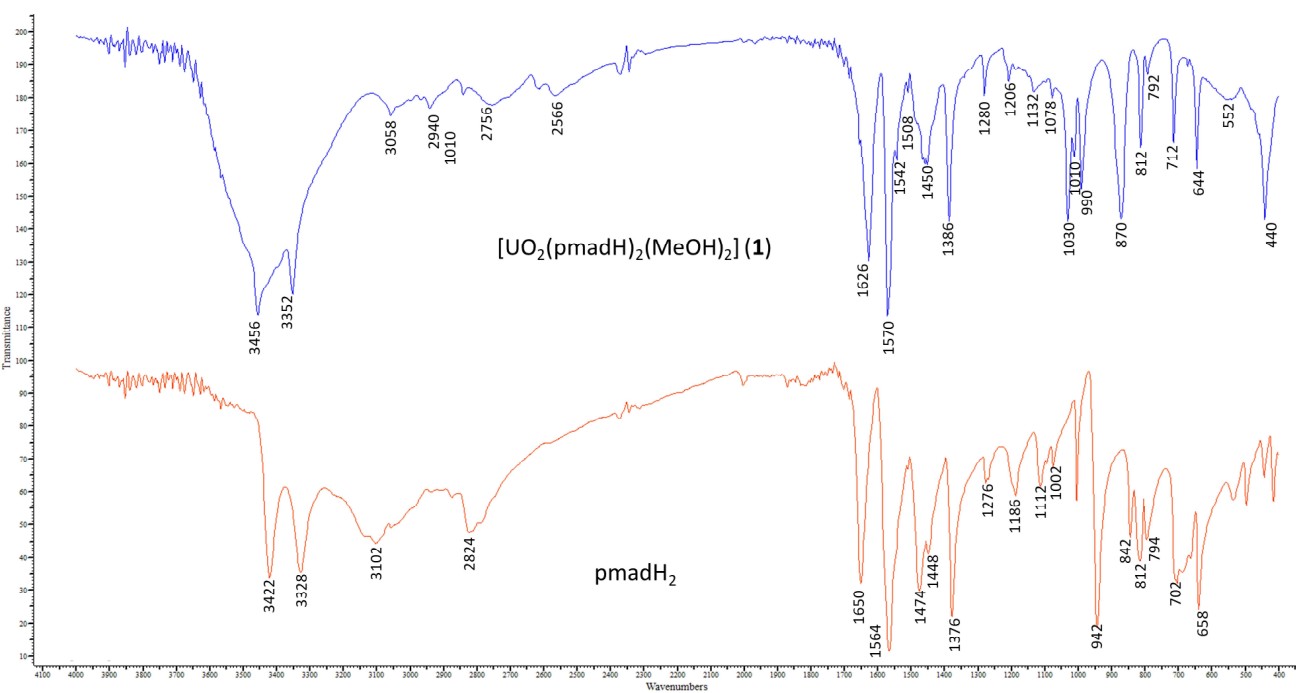

**Figure 6.** The IR spectra (KBr, cm$^{-1}$) of pmadH$_2$ and [UO$_2$(pmadH)$_2$(MeOH)$_2$] (**1**).

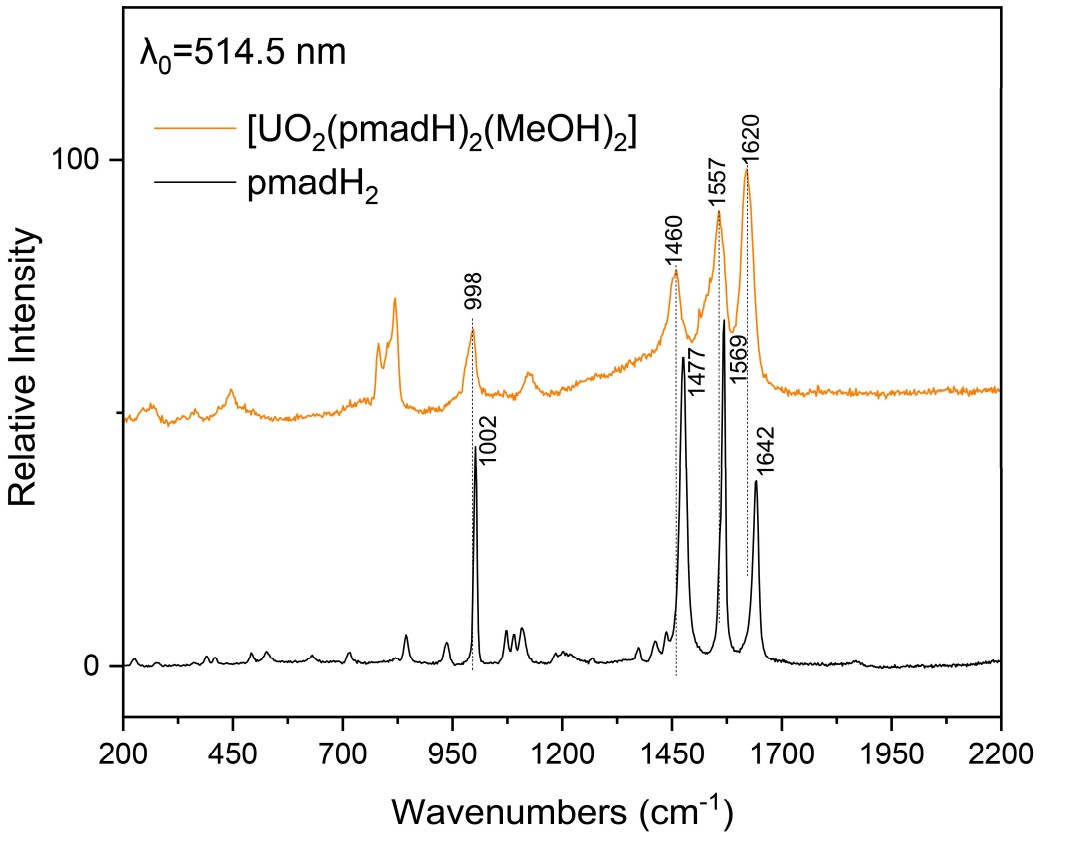

**Figure 7.** Partially labeled Raman spectra of pmadH$_2$ (bottom) and [UO$_2$(pmadH)$_2$(MeOH)$_2$ (**1**) (up) in the 2200–200 cm$^{-1}$ region.

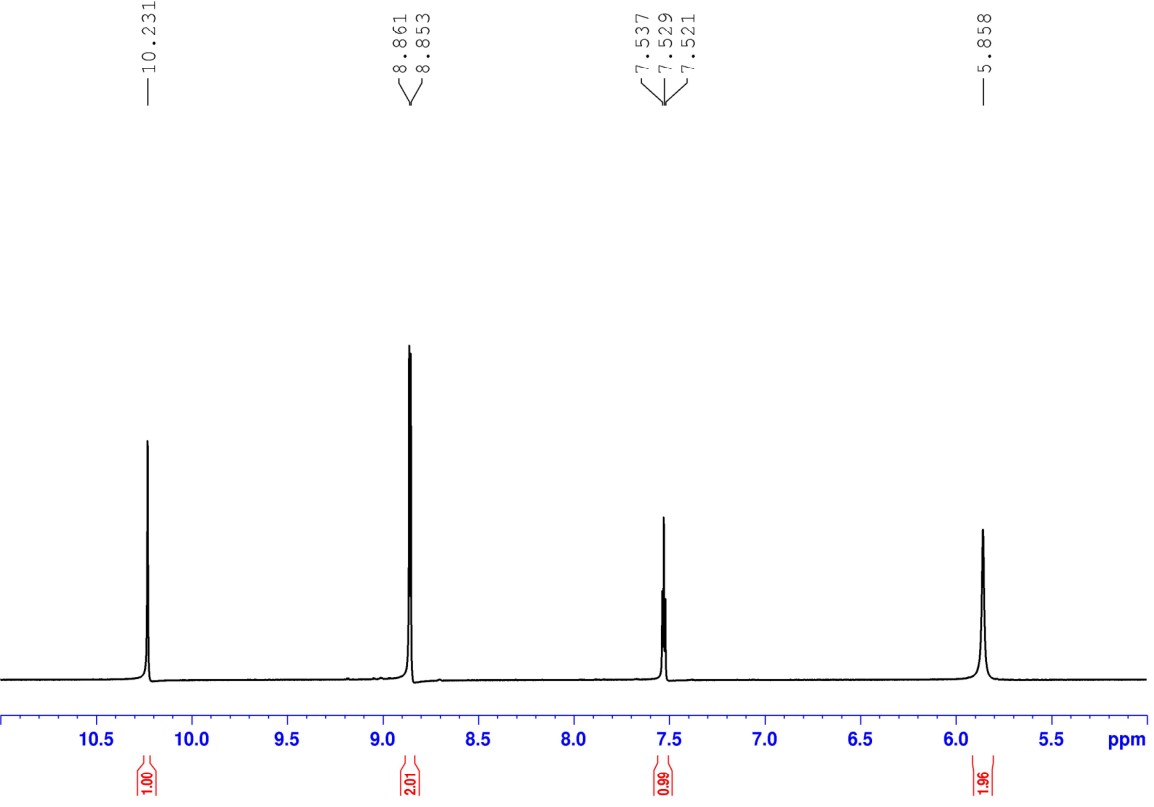

**Figure 8.** The $^1$H NMR spectrum of [UO$_2$(pmadH)$_2$(MeOH)$_2$] (**1**) in d$_6$-DMSO in the $\delta$ 11.00–5.00 ppm region.

Complex **1** is soluble in DMSO, and attempts were made to study its behavior in this solvent. The color of the solutions is reddish-orange, somewhat different from the color of the solid. The UV region is dominated by a strong absorption band at 289 nm, assigned to a $\pi\rightarrow\pi^*$ transition of the organic ligand. To our surprise, the spectrum of free pmadH$_2$ exhibits a band at exactly the same wavelength; a shift was expected in the spectrum of **1** due to deprotonation (pmadH$^-$) and coordination. This striking similarity is the first evidence of the decomposition of the complex in DMSO. A weak band at ∼515 nm is also located in the spectrum of the complex. Uranyl complexes typically absorb at ∼430 nm. This band is responsible for the yellow color of most uranyl complexes [56], and it is due to ligand-to-metal charge transfer (LMCT) transition. In the present case, the LMCT transition is red-shifted by ∼85 nm, resulting in the band at ∼515 nm, which gives the reddish-orange color of the solution. The appearance of such a band is evidence for the weakening of the U$^{VI}$-O$_{yl}$ bonds (observed in the structure of **1**) and the bending of the *trans*-{O=U=O}$^{2+}$ (not observed in the solid-state structure of **1**) [56].

To further probe the solution behavior of **1**, we examined first the conductivity of DMSO solutions at 25 °C using various concentrations. The solutions are non-conductive, suggesting the presence of neutral species. The value of molar conductivity $\Lambda_M$ (DMSO, $10^{-3}$ M, 25 °C) is 6 S cm$^2$ mol$^{-1}$, confirming that the complex is non-electrolyte [57]; note that, for the determination of $\Lambda_M$, we assumed that the molecular weight of **1** is that of the solid complex (which might be wrong; *vide infra*).

In the second step, we recorded the $^1$H NMR spectra of pmadH$_2$ and **1** in d$_6$-DMSO. The spectra are completely identical, except for the appearance of the MeOH signals in the latter. The spectra show a singlet signal at $\delta$ 10.23 ppm, attributed to the -OH group; a multiplet signal at $\delta$ 8.85 ppm due to the two aromatic protons next to the pyrimidine-ring N atoms; a triplet at $\delta$ 7.53 assigned to the remaining aromatic proton and a somewhat broad; and a singlet signal at $\delta$ 5.83 ppm due to the -NH$_2$ protons [35]. The integration ratio of the four signals is perfectly 1:2:1:2, as expected. Additionally, the $^1$H NMR spectrum of **1**

exhibits a doublet at $\delta$ 3.19 ppm (well separated from the signal of $H_2O$ contained in the deuterated solvent) attributed to the methyl protons and a quadruple signal at $\delta$ 4.22 ppm ascribed to the hydroxyl group of MeOH [58]. The integration ratio of the signals at $\delta$ 10.23, 8.85, 7.53, 5.83, 4.22, and 3.19 ppm is 1:2:1:2:1:3, indicating a 1:1 (or 2:2) $pmadH_2$:MeOH ratio in solution.

The above-described UV/Vis, molar conductivity, and [1] NMR data for **1** (when interpreted in a combined way) clearly demonstrate that the complex decomposes in DMSO, and there are no $[UO_2(pmadH)_2(MeOH)_2]$ molecules in solution. The ligand $pmadH^-$ is protonated ([1]H NMR evidence), giving free, i.e., uncoordinated, $pmadH_2$ molecules. The source of the protons cannot be MeOH because this is in its neutral form ([1]H NMR evidence), and furthermore, the $pmadH_2$:MeOH molar ratio in solution is 1:1 (or 2:2), identical to the $pmadH^-$:MeOH ratio in the solid complex. Thus, we tentatively propose that the proton source is the $H_2O$ contained in DMSO (Equations (4)–(6)). It should be mentioned at this point that the deuterated DMSO used for the [1]H NMR spectra and the commercially available DMSO used for the conductivity measurements and UV/Vis spectra both contain $H_2O$. Thus, we strongly believe that the decomposition is due to $H_2O$. Experiments with absolute DMSO were not performed.

$$[UO_2(pmadH)_2(MeOH)_2] + 2H_2O \rightarrow [UO_2(pmadH)_2(H_2O)_2] + 2MeOH \quad (4)$$

$$[UO_2(pmadH)_2(H_2O)_2] + 2H_2O \rightarrow [UO_2(pmadH)_2(OH)_2]^{2-} + 2H_3O^+ \quad (5)$$

$$[UO_2(pmadH)_2(OH)_2]^{2-} + 2H_3O^+ + 2H_2O \rightarrow\rightarrow [UO_2(H_2O)_4(OH)_2] + 2pmadH_2 \quad (6)$$

## 3. Materials and Methods

### 3.1. Materials and Instrumentation

All manipulations were performed under aerobic conditions. Reagents and solvents were purchased from Alfa Aesar and Sigma-Aldrich and were used as received. The deuteration degree of $d_6$-DMSO was 99.8%, with the rest being mainly $H_2O$. The actual $H_2O$ content was certainly higher because the solvent is hygroscopic, and the syringe used had not been dried under Ar. The $H_2O$ content is also present in the commercially available DMSO used for the UV/Vis spectra and conductivity measurements (1–2%). The compound pyrimidine-2-amidoxime ($pmadH_2$) was synthesized through a 2:2:1 reaction between 2-cyanopyrimidine, $NH_2OH \cdot HCl$, and $Na_2CO_3$ in $H_2O$/EtOH under reflux, as reported earlier [49]; the yield was 75%. Its purity was assessed using [1]H NMR spectroscopy in $d_6$-DMSO [42]. Deionized water was obtained from the in-house facility. ***Safety note***: Compounds with high nitrogen content (such as $pmadH_2$) are potentially explosive, and caution is recommended when handling them, e.g., using a plastic spatula. With the small quantities in the present work, we did not encounter any problems. ***Warning***: The uranyl salts used in this work are highly toxic, can affect breathing, and may be absorbed through the skin. Contact can irritate and burn the skin and eyes with possible eye damage. In terms of breathing, it can irritate the nose, throat, and lungs, causing coughing, wheezing, and/or shortness of breath. Although the depleted uranium used in this work has a very long half-life, precautions for working with radioactive substances must be strictly followed. All reactions were carried out in a fume hood containing $a$- and $\beta$-counting equipment, and all manipulations were carried out using masks and gloves.

C, H, and N microanalyses were conducted at the Instrumental Analysis Center of the University of Patras. Conductivity measurements were carried out at $25 \pm 1$ °C with a Metrohm–Herisau E-527 bridge and a cell of standard design. The concentration was $\sim 10^{-3}$ M, assuming the formula weight of compound **1**. FT-IR spectra (4000–400 cm$^{-1}$) were recorded using a Perkin-Elmer 16PC spectrometer. The samples were in the form of KBr pellets. Sometimes, attenuated total reflectance (ATR) spectra were recorded on a Bruker Optics Alpha-P Diamond ATR spectrometer. Raman spectra were recorded on a

T-64000 Jobin Yvon-Horiba micro-Raman setup, utilized in a single-spectrograph configuration. The excitation wavelength was 514.5 nm emitted from a DPSS laser (Cobolt Fandango TMISO laser). The collected backscattered radiation was directed to the monochromator (single configuration) and a Spectraview-2DTM liquid $N_2$-cooled CCD detector. The laser was focused on the samples using a $50\times$ microscope objective with a power of 2 mW on the samples. The resolution was kept constant at $\sim4$ cm$^{-1}$ in all experiments. UV/Vis spectra in DMSO were recorded on a Hitachi U-3000 spectrometer. The sample concentration was 100 μg/mL. $^1$H NMR spectra in d$_6$-DMSO were run on a 600.13 MHz Bruker Avance DPX spectrometer.

*3.2. Preparation of the Complex*

Method (a): Solid $UO_2(NO_3)_2 \cdot 6H_2O$ (0.050 g, 0.10 mmol), pmadH$_2$ (0.028, 0.20 mmol), and NaOMe (0.011 g, 0.20 mmol) were added to a solvent mixture of MeOH/MeCN (20 mL, 1:1 $v/v$). The resulting yellow suspension was stirred for 1 h, and the yellowish-orange solid was collected through filtration, washed with cold MeCN (1 mL) and Et$_2$O (2 × 2 mL), and dried in air. Yield: 42%. Anal. Calcd. (%) for $C_{12}H_{18}N_8O_6U$: C, 23.69; H 2.99; N, 18.42. Found (%): C, 23.87; H, 3.06; N, 17.66. $\Lambda_M$ (DMSO, 25 °C, $10^{-3}$ M)= 6 S cm$^2$ mol$^{-1}$. IR (KBr, cm$^{-1}$): 3456 s, 3352 s, $\sim$3250 sh, 3058 w, 2940 w, 1626 s, 1570 s, 1542 sh, 1508 sh, 1450 mb, 1386 s, 1280 w, 1206 w, 1132 w, 1078 w, 1030 s, 1010 sh, 990 m, 870 s, 812 m, 792 w, 712 m, 644 m, 552 wb, 440 s. Selected Raman peaks (cm$^{-1}$): 1620 s, 1557 s, 1512 w, 1460 m, 998 m, 820 s, 782 m, 445 w, 262 w. UV/Vis (DMSO): 289 sb, 515 w. $^1$H NMR (d$_6$-DMSO, $\delta$/ppm): 10.23 (s, 1H), 8.85 (mt, 2H), 7.53 (t, 1H), 5.83 (s/b, 2H), 4.22 (q, 1H), $\sim$3.5 (see text), 3.19 (d, 3H). The coordinated hydroxide protons (see Equation (6)) are most likely hidden under the very broad $H_2O$ signal at $\delta \sim$3.5 ppm [59] (Figure S4). The same product in a microcrystalline powdered form would be obtained (yield ca. 50%) if, to a solution of pmadH$_2$ in MeCN/MeOH, solid NaOMe was first added and then solid $UO_2(NO_3)_2 \cdot 6H_2O$ (same quantities and volume as above). The yellowish-orange solid was precipitated after 10 min of stirring, which was further continued for 50 min before isolation. The orange crystals of the product, suitable for X-ray crystallography, were obtained from the filtrate of the above reaction mixtures (without Et$_2$O and MeOH used for the washings) after storage in a closed vial at 25 °C for 24 h. The crystals were washed with Et$_2$O (2 × 1 mL), dried in air, and weighed. The total yield (powder plus crystals) approached $\sim$60%. The IR and $^1$H NMR spectra of the powdered crystals are identical to the corresponding spectra of the initially obtained powder. To further prove that the powder and the crystals represent the same material, the powdered crystals underwent microanalyses. Anal. Calcd. (%) for $C_{12}H_{18}N_8O_6U$: C, 23.69; H, 2.99; N, 18.42. Found (%): C, 23.37; H, 3.09; N, 17.89.

Method (b): A yellow solution of $UO_2(NO_3)_2 \cdot 6H_2O$ (0.050 g, 0.10 mmol) in MeOH/MeCN (10 mL, 1:1 $v/v$) was slowly added to a solution of pmadH$_2$ (0.028 g, 0.20 mmol) and Et$_3$N (28 μL, 0.20 mmol) in MeOH/MeCN (10 mL, 1:1 $v/v$). The color of the solution turned to orange, and upon vigorous stirring for 5 min, an orange solid was precipitated. The stirring continued for an additional 55 min, and the precipitate was collected through filtration, washed with cold MeOH (1 mL) and Et$_2$O (2 × 2 mL), and dried in air. Yield: 29%, The IR and $^1$H NMR spectra of the sample are identical to the corresponding spectra of authentic **1** prepared using method (a).

Method (c): To a stirred yellow solution of $UO_2(O_2CMe)_2 \cdot 2H_2O$ (0.042 g, 0.10 mmol) in warm MeOH (7 mL), a solution containing pmadH$_2$ (0.028 g, 0.20 mmol) and NaOMe (0.011 g, 0.20 mmol) in MeOH/MeCN (10 mL, 1:3 $v/v$) was added. The resulting solution immediately turned orange, and a solid of the same color was precipitated after stirring for ca. 5 min. The stirring continued for an additional 55 min, and the orange solid was isolated as in method (b). The IR spectrum of the sample is identical to the corresponding spectrum of authentic **1** prepared using method (a). $^1$H NMR (d$_6$-DMSO, $\delta$/ppm): 10.25 (s, 1H), 8.84 (mt, 2H), 7.54 (t, 1H), 5.80 (s/b, 2H), 4.23 (q, 1H), $\sim$3.5 [due to the $H_2O$ signal that hides the coordinated hydroxo-protons; see Equation (6)], 3.18 (d, 3H).

*3.3. Single-Crystal X-ray Crystallography*

An orange crystal of **1** (0.11 × 0.16 × 0.18 mm) was taken directly from the mother liquor and immediately cooled to −93 °C. Diffraction data were collected on a Rigaku R-Axis Image Plate diffractometer using graphite-monochromated Mo Kα radiation. Data collection (ω scans) and processing (cell refinement, data reduction, and empirical/numerical absorption) were performed using the CrystalClear program package [60]. The structure was solved with direct methods using SHELXS, ver. 2013/1 [61] and refined using full-matrix least-square techniques on $F^2$ with SHELXL, ver. 2014/6 [62]. H atoms were either located using difference maps and refined isotropically or introduced at calculated positions and refined as riding on their corresponding bonded atoms. All non-H atoms were refined anisotropically. Plots of the structures were drawn using the Diamond 3 program package [63]. Further crystallographic details of **1** not listed in Table 1: $2\theta_{max} = 54.0°$, $[\Delta/\sigma]_{max} = 0.001$, $R_1/wR_2$ (for all data) = 0.0136/0.332.

Crystallographic data were submitted to the Cambridge Crystallographic Data Center, No. 2331640. Copies of the data can be obtained free of charge upon application to CCDC, 12 Union Road, Cambridge, CB2 1EZ, UK. Telephone: +(44)-1223-336033; E-mail: deposit@ccdc.ac.uk, or via https://www.ccdc.cam.ac.uk/structures/.

## 4. Concluding Comments and Perspectives

In this work, we reported the employment of pyrimidine-2-amidoxime (pmdaH$_2$) in reactions with uranyl sources. Although there are probably additional products from the general *trans*-{U$^{VI}$O$_2$}$^{2+}$/pmadH$_2$ reaction system (see Section 2.1), we could crystallize (and hence structurally characterize) and fully study only complex [UO$_2$(pmadH$_2$)(MeOH)$_2$] (**1**), obtained from the presence of an external base in the reaction mixture. From the chemistry viewpoint, the most salient features of this work are as follows: (a) Compound **1** is the first structurally characterized 5f metal complex of pmadH$_2$ and, in general, of the ligands shown in Figure 2; (b) the complex exhibits interesting structural features including the ligation mode 1.11000 (Figure 5) observed for the first time in the coordination chemistry of pmadH$_2$ and the formation of two stable three-membered chelating rings per U$^{VI}$ atom; this motif is very rare [16,18,22,23] in uranyl complexes with other amidoxime ligands; and (c) the three-membered amidoximate UÔN chelating ring appears unstable in solution (at least in water-containing DMSO).

From the technological viewpoint, which stimulated our efforts, the present results should not be exaggerated. The most reliable methodology to answer the questions concerning the selective extraction of *trans*-{U$^{VI}$O$_2$}$^{2+}$ from the oceans with amidoxime-functionalized materials remains the exposure of polymeric fibers into environmental seawater and a subsequent thorough study of the extracted solid. Our efforts in this work again confirm the thermodynamic stability of the three-membered chelating ring when the amidoxime group is singly deprotonated (this is the case in the pH of seawater). The ease of formation of this ring is widely considered [8,27] as the key to the strong binding and selectivity of the uranyl ion with amidoxime materials. However, the decomposition of the complex in DMSO that contains H$_2$O indicates that the three-membered ring might be unstable in water (at least with the present ligand); another source of H$_2$O is the starting uranyl material, which is hydrated. This means that (i) ligands should be designed that form stable complexes with the uranyl ion in water at approximately neutral pH, and (ii) the formation of the three-membered ring seems unstable in aqueous media. Moreover, the coordination mode of pmadH$^-$ in **1** demonstrates that the influence of neighboring (to the amidoxime functionality) donor sites might not be important. One of the heterocyclic N atoms of pmadH$^-$ could easily form a stable five-membered chelating ring with the participation of the oximate N, leaving the deprotonated O available for bridging one or two uranyl ions. This is the case with other metal ions (see Figure 5), but the uranyl ion does not follow this trend, and it thus presents a unique behavior.

With the knowledge and experience obtained through the present study, our efforts will be directed toward the synthesis and characterization of uranyl complexes with other

amidoxime ligands, the investigation of whether the three-membered chelating ring is stable or unstable (as in **1**) in aqueous media, the study of the reactions of the ligands with $\{V^VO_2\}^+$ (the main competitor for the recovery of $\{UO_2\}^{2+}$ from seawater), and the discovery of synthetic protocols for the preparation of uranyl complexes with three equatorial three-membered amidoxime chelating rings per metal ion [28].

**Supplementary Materials:** The following supporting information can be downloaded at: https://www.mdpi.com/article/10.3390/inorganics12030082/s1, Figure S1: The packing of molecules in the crystal structure of complex **1** viewed along the [1, −1, 1] direction; Figure S2: The UV/Vis spectrum of complex **1** in DMSO; Figure S3: $^1$H NMR spectrum of the free ligand pmadH$_2$ in d$_6$-DMSO in the $\delta$ 11.00–5.00 ppm region; Figure S4: A part of the $^1$H NMR spectrum of complex **1** in d$_6$-DMSO showing the MeOH protons.

**Author Contributions:** S.T.T. and D.I.T. synthesized and proved the purity of the free ligand pmadH$_2$ and performed a comprehensive literature search. S.T.T. and S.G.S. contributed to the synthesis, crystallization, and IR and $^1$H NMR characterization of the complex. Z.G.L. recorded the UV/Vis and Raman spectra of the compounds and interpreted the results. C.P.R. and V.P. collected single-crystal X-ray data and solved the structure; the latter also studied the supramolecular features of the reported structure and wrote the relevant part of the paper. S.T.T. and S.P.P. coordinated the research; the latter wrote the article based on the detailed reports of the collaborators. All authors exchanged opinions concerning the progress of the experiments and commented on the various drafts of the paper. All authors have read and agreed to the published version of the manuscript.

**Funding:** C.P.R. and V.P. thank the Special Account of the NCSR "Demokritos" for financial support concerning the operation of the X-ray diffractometer at INN through the internal program titled "Structural Study and Characterization of Crystalline Materials" (NCSR "Demokritos", ELKE ≠10813).

**Data Availability Statement:** Data are contained within the article and Supplementary Materials.

**Acknowledgments:** The authors are grateful to the Head of the Laboratory of Applied Molecular Spectroscopy Research, George A. Voyiatzis (ICE-HT/FORTH), for the access to the Raman facilities. S.P.P and S.T.T. also thank Spyros N. Yannopoulos for the purchase of few chemicals used.

**Conflicts of Interest:** The authors declare no conflicts of interest.

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

Metalloligand (H₂L=pyrimidine-2-carboxamide oxime): A Theoretical and Experimental Magneto-Structural Study. *Eur. J. Inorg. Chem.* **2011**, *2011*, 5225–5232. [CrossRef]

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
