# Peer review of "Understanding the Selective Extraction of the Uranyl Ion from Seawater with Amidoxime-Functionalized Materials: Uranyl Complexes of Pyrimidine-2-amidoxime†"

_inorganics, doi:10.3390/inorganics12030082_

Round 1

Reviewer 1 Report

Comments and Suggestions for Authors

Generally, the experimental findings of this work are very interesting and carried out carefully and therefore worth publishing. However, the context and writing in general sometimes seems a little strange and needs major improvements. It sounds like an interim project report and not like a manuscript for a peer-review journal. Specific comments are listed below.

Title: The term “Modeling” is misleading; one expects theoretical calculations for modeling.

Abstract: Too long, too general. The first 5 introductory sentences could be summarized to 1 or max. 2. Mentioning explicitly VO2 in the abstract should be avoided, one would expect comparable experiments with this cation.

Introduction: The introduction is generally too verbose in terms of content.

The first half of the first paragraph (up to “…discovery of the Stokes shift and radioactivity.”) could be deleted.

Second paragraph: a reference should be given for the amount of dissolved U in the oceans.

Fourth paragraph (starting with “Based on the above mentioned information, and as a continuation of our interest in various aspects of actinide chemistry…”) sounds like a mixture between project proposal and interim project report:

(i)                  It is not necessary in a paper to document all work which is done in the working group (i.e. “… various aspects of actinoid chemistry” and the cited own references).

(ii)                It is not necessary to list all aspects of the whole project (long-term goals (a) … (f)). In a paper, a separate, well-rounded, self-contained topic should be presented. Of course it can be embedded in a larger context, but this should only be described briefly.

(iii)               “The present work is a preliminary contribution to…” – “Preliminary" results should not be presented in a peer-review article. This belongs in an internal interim project report. The results in a peer-reviewed article should be solid and conclusively verified.

(iv)               “This paper is the first one of a series of articles concerning the reactions…” Such a statement should be avoided (Don't cross your bridges before you come to them.). – This sentence can be deleted.

Figure 2 is not necessary, because only the ligand pmadH2 is used.

The last sentence of the introduction (“the present work can also be considered as a continuation…”) can be deleted.

A short statement about what will be presented and which methods were used should be at the end of the introduction.

Results and discussion:

The heading 2.1. “Synthetic comments” should be rephrased to “Synthesis of …”

Instead of “For reasons outlined in Introduction” the reasons should be stated briefly, concisely and precisely.

“Hundreds of efforts” does not sound very scientific…

The heading 2.2. ”Description of structure” should be rephrased to “Structure determination of…”

Spectroscopic characterization:

-          IR spectra of both, ligand and UO2-complex, should be shown in one Figure for better comparison and should appear in the main text, not in SI (with clever and readable labelling to clearly demonstrate the shifts, or possibly a Table could be helpful)

-          The Raman spectra needs clever labelling to clearly demonstrate the shifts (again, a Table could be helpful).

-          The UV/vis spectrum can be shifted into the supporting information, because there is no difference between ligand and complex.

-          The description of the decomposition of the complex is too long and detailed and generally questionable for 2 reasons: (i) the purity of DMSO is not specified. Shouldn’t it be almost anhydrous? - Experiments with "clean" DMSO would be helpful. (ii) Isn’t it necessary to use ligands that form stable complexes with uranium both in water and in the organic phase to extract uranium from seawater? - The results should at least be put into a better context by considering these ideas.

Materials and methods:

-          The puritiy grade of DMSO should be mentioned.

Concluding comments

-          “It is difficult to conclude on a work that is still at its infancy.” This sentence is unfortunately worded. Of course you can also draw conclusions from pioneering work (you demonstrate it with the conclusion section). - Best to leave it out.

-          Last paragraph: The list of future work and planned publications is too detailed ( this sounds again like an internal interim project report).

Author Response

Generally, the experimental findings of this work are very interesting and carried out carefully and therefore worth publishing. However, the context and writing in general sometimes seems a little strange and needs major improvements. It sounds like an interim project report and not like a manuscript for a peer-review journal. Specific comments are listed below.

We thank Reviewer #1 for her/his time to study the ms and the valuable comments provided. We are glad to see that she/he believes that our ms is ‘’worthy publishing’’. We have taken into account almost ALL of the revision points/comments/suggestions.

Title: The term “Modeling” is misleading; one expects theoretical calculations for modeling.

We have changed the term ‘’Modeling’’ with the term ‘’Understanding’’.

Abstract: Too long, too general. The first 5 introductory sentences could be summarized to 1 or max. 2. Mentioning explicitly VO2 in the abstract should be avoided, one would expect comparable experiments with this cation.

The comment is correct. We have removed three (3) out of five (5) introductory sentences; we also avoided to mention {VVO2}2+.

Introduction: The introduction is generally too verbose in terms of content.

We agree. The ‘’introduction’’ has been significantly condensed.

The first half of the first paragraph (up to “…discovery of the Stokes shift and radioactivity.”) could be deleted.

The first half of the first paragraph has been deleted in the revised ms, as suggested. As a result, the former ref.[1], which is now irrelevant, has been removed.

Second paragraph: a reference should be given for the amount of dissolved U in the oceans.

The appropriate reference is given in the revised version of the ms.

Fourth paragraph (starting with “Based on the above mentioned information, and as a continuation of our interest in various aspects of actinide chemistry…”) sounds like a mixture between project proposal and interim project report:

(i) It is not necessary in a paper to document all work which is done in the working group (i.e. “… various aspects of actinoid chemistry” and the cited own references).

(ii) It is not necessary to list all aspects of the whole project (long-term goals (a) … (f)). In a paper, a separate, well-rounded, self-contained topic should be presented. Of course it can be embedded in a larger context, but this should only be described briefly.

(iii) “The present work is a preliminary contribution to…” – “Preliminary" results should not be presented in a peer-review article. This belongs in an internal interim project report. The results in a peer-reviewed article should be solid and conclusively verified.

(iv) “This paper is the first one of a series of articles concerning the reactions…” Such a statement should be avoided (Don't cross your bridges before you come to them.). – This sentence can be deleted.

The comments and suggestions are correct. In detail:

  1. We have deleted the relevant sentence and the citations of our references (refs. 33-35 of the initially submitted ms).
  2. As suggested, we have deleted the list of many aspects of the whole project (long terms…). We briefly describe the goals, embedded in the main context.
  • We have deleted the sentence about the ‘’preliminary contribution’’.
  1. We have omitted the relevant sentence.

Figure 2 is not necessary, because only the ligand pmadH2 is used.

We have removed the structural formulae of all other ligands, except the structural formula of the ligand used in the present work. Thus, the figure is retained but it contains only pmadH2; we have not completely deleted the whole figure because we believe that the representation of the ligand is helpful for the non-familiar readers.

The last sentence of the introduction (“the present work can also be considered as a continuation…”) can be deleted.

The sentence has been deleted.

A short statement about what will be presented and which methods were used should be at the end of the introduction.

The short statement has been added at the end of the ‘’Introduction’’ section of the revised ms.

The heading 2.1. “Synthetic comments” should be rephrased to “Synthesis of …”

The heading 2.1 has been rephrased to ‘’Synthesis of the Complex’’.

Instead of “For reasons outlined in Introduction” the reasons should be stated briefly, concisely and precisely.

The reasons are now stated briefly, concisely and precisely. We have removed the sentence ‘’For reasons outlined in the ‘Introduction’… system’’.

“Hundreds of efforts” does not sound very scientific…

We agree and have rephrased the relevant sentence.

Results and discussion:

The heading 2.2. ”Description of structure” should be rephrased to “Structure determination of…”

The heading 2.2 has been rephrased to ‘’Structure Determination of the Complex’’.

IR spectra of both, ligand and UO2-complex, should be shown in one Figure for better comparison and should appear in the main text, not in SI (with clever and readable labelling to clearly demonstrate the shifts, or possibly a Table could be helpful).

We have exactly followed the Reviewer’s suggestion. Figure 6 of the revised main ms shows the labelled IR spectrum of both the free ligand and the complex. The coordination shifts are discussed in the text, and we believe that there is no need for a table. As a consequence, Figures S2 and S3 have been removed from SI.

The Raman spectra needs clever labelling to clearly demonstrate the shifts (again, a Table could be helpful).

The main peaks of the Raman spectra have been labelled in the revised Figure 7. Coordination shifts are briefly mentioned in the text, and we believe there is no need for a table.

The UV/vis spectrum can be shifted into the supporting information, because there is no difference between ligand and complex.

The UV/Vis spectrum has been now moved into the Supporting Information and it is now Figure S2.

The description of the decomposition of the complex is too long and detailed and generally questionable for 2 reasons: (i) the purity of DMSO is not specified. Shouldn’t it be almost anhydrous? - Experiments with "clean" DMSO would be helpful. (ii) Isn’t it necessary to use ligands that form stable complexes with uranium both in water and in the organic phase to extract uranium from seawater? - The results should at least be put into a better context by considering these ideas.

First, the description of the decomposition of the complex has been drastically condensed, as (correctly) suggested by the Reviewer. (i) We have specified the purity of d6-DMSO and commercially available DMSO (used for UV/Vis spectra and conductivity measurements) in Part 3.1 of the ‘’Materials and Methods’’ section. Certainly, both contain H2O. This is also briefly mention in Part 3.2 (‘’Spectroscopic Characterization). We agree that experiments with ‘’clean’’ DMSO would be helpful, but such studies have not been performed and this is clearly (and honestly) stated in this part of the revised ms. Thus, there is strong evidence that H2O (also present in the starting uranyl material) causes decomposition of the complex in DMSO. (ii) The comment is correct and it is, indeed, necessary to use ligands that form stable complexes with the uranyl ion in water. This comment and the necessity for new amidoxime ligands, that form stable complexes in water, has been added in the ‘’Concluding Comments and Perspectives’’ of the revised ms. Concerning the Reviewer’s comment that the complex should be stable in organic solvents, we do believe that it is not a prerequisite because the recovery of the uranyl ion form seawater does not involve organic phase extraction, but it is a process in which uranium is adsorbed on water-insoluble polymers functionalized with the amidoxime group in aqueous environments.

Materials and methods:

  • The purity grade of DMSO should be mentioned.

This is mentioned in both Parts 2.3 and 3.1, please see our answer above.

Concluding comments

- “It is difficult to conclude on a work that is still at its infancy.” This sentence is unfortunately worded. Of course you can also draw conclusions from pioneering work (you demonstrate it with the conclusion section). - Best to leave it out.

The sentence has been left out, as suggested.

- Last paragraph: The list of future work and planned publications is too detailed ( this sounds again like an internal interim project report).

The last paragraph has been condensed and reorganized, avoiding plans for future submissions, etc.

In summary, we are grateful to Reviewer #1 because her/his exhaustive and scientifically solid review has led to a significant improvement of the quality of the ms.

Reviewer 2 Report

Comments and Suggestions for Authors

This article discussed the selective extraction of the uranyl ion from seawater using amidoxime-functionalized materials. The study focused on the ligation modes and binding strength of the amidoxime group with the uranyl ion and other competing metal ions. The research presents results on the reactions of uranyl and pyrimidine-2-amidoxime, providing insights into the complex structure and interactions. Spectroscopic techniques confirm the decomposition of the complex in DMSO, releasing neutral pmadH2 molecules. The study aims to enhance the recovery capacity of uranium from seawater using bifunctional ligands and polymeric sorbents. The basic logic of the article is clear and the writing is fluent, but there are still some questions could be handled before official publication.

1. It would be better to check the spelling and usage of English words and sentences in this article.

2. It would be better to provide the result of 1H NMR spectra of the sample prepared through method (c), and compare with authentic 1.

3. Could the crystal data of 1 be compared well with the Cambridge Crystallographic Data Center?

4. There is a lack of sufficient experiments on the performance of the material for uranium. What about the experimental data of the selectivity?

5. Have you conducted simulated seawater experiments, which could be useful for reflecting the practical value of the material?

6. Fig. 7, what does the ghost peak mean, did you compare the spectrum of blank as control.

Comments on the Quality of English Language

Minor editing of English language required.

Author Response

This article discussed the selective extraction of the uranyl ion from seawater using amidoxime-functionalized materials. The study focused on the ligation modes and binding strength of the amidoxime group with the uranyl ion and other competing metal ions. The research presents results on the reactions of uranyl and pyrimidine-2-amidoxime, providing insights into the complex structure and interactions. Spectroscopic techniques confirm the decomposition of the complex in DMSO, releasing neutral pmadH2 molecules. The study aims to enhance the recovery capacity of uranium from seawater using bifunctional ligands and polymeric sorbents. The basic logic of the article is clear and the writing is fluent, but there are still some questions could be handled before official publication.

We thank Reviewer #2 for her/his time to study the ms and the valuable comments. We are glad to read her/his warm words concerning the scientific content, organization and writing of the article. The comments/suggestions are clear and scientifically solid. We have been able to address MANY (but not all) the revision points/comments/suggestions raised by the Reviewer. In detail:

  1. It would be better to check the spelling and usage of English words and sentences in this article.

We went carefully through the article, and improved the spelling usage of English words and sentences.

  1. It would be better to provide the result of 1H NMR spectra of the sample prepared through method (c), and compare with authentic 1.

We have added the results of the 1H NMR spectrum of the sample prepared through method (c); as it is clear, beyond any doubt, the spectrum is identical with that prepared through method (a) whose δ values are also listed in Part 3.2 of section 3 (‘’Materials and Methods’’). We avoid to incorporate the spectrum of the former [method (c)] in Figure 8 [ which refers to the sample prepared by method (a)], because the two spectra are identical.

  1. Could the crystal data of 1 be compared well with the Cambridge Crystallographic Data Center?

Carefully inspection of the CCDC reveals that 1 is a new compound, i.e. it is the first structurally characterized uranium complex with any form (neutral, monoanionic, dianionic) form of the ligand pmadH2. All the other, previously structurally characterized metal complexes of pmadH2/pmadH-/pmad2- had already been listed in Table 4 of the initially submitted ms, and this table remains in the revised version of the ms.

  1. There is a lack of sufficient experiments on the performance of the material for uranium. What about the experimental data of the selectivity? Have you conducted simulated seawater experiments, which could be useful for reflecting the practical value of the material?

Both questions are logical! The answer in both question is ‘’No’’. Such studies, which are beyond of the scope of this work, require the cooperation with polymer scientists who can copolymerize polyethylene or propylene fibers with side chains composed of pmadH2-based polynitrile by the RIGP (radiation-induced grafting polymerization) method and subsequently convert he nitrile groups through reaction with H2NOH (a good solvent mixture for this conversion would be water/ethanol), please see Figure 1 of the main text; these materials could be tested for the recovery and selectivity of the uranyl ion using simulated seawater. Such studies may be goals for the future, but first we should prepare and characterize uranyl complexes with a variety of amidoxime ligands. The work here is a coordination chemistry approach of the problem. Our short-term goals are: (a) the full investigation and understanding (in combination with DFT studies) of the uranyl-amidoxime interactions; (b) the creation of synthetic models for the highly selective removal of uranyl from seawater with amidoxime-functionalized materials; (c) the investigation of the way in which a pair of amidoxime groups can coordinate to the uranyl ion (this requires synthesis of the suitable ligands); (d) the design of bifunctional ligands containing amidoxime groups and a neighboring donor site, and their reactions with uranyl sources, in order to understand the reactivity of polymeric sorbents possessing -in addition to the amidoxime group- a second, different donor site (e.g. a carboxylate group) on the side chains; (e) structural comparison of {UO2}2+ and {VO2}+ (the main competitor in the recovery of uranium from seawater) complexes with amidoxime ligands. We have been working in the implementation of the above five goals, and all our studies are based on small molecules; the present work is a contribution to some aspects of the goals a,b, and d. Our long-term goal, which requires the experiments mentioned by Reviewer #2, is the proposal of more efficient amidoxime-containing high surface-area polymeric fiber adsorbents.

  1. Figure 7, what does the ghost peak mean, did you compare the spectrum of blank as control.

First of all, we inform Reviewer #2 that Figure 7 has been transferred to the Supporting Information (it is now Figure S2), upon request of Reviewer #1. Concerning the correct question by Reviewer #1, the ghost peak is due to a known (to us) technical issue of the instrument; it appears even if we compare the spectrum of the blank as control.

In summary, the comments by Reviewer #2 have been assisted in the improvement of quality of our ms, and we are grateful to her/him.

Round 2

Reviewer 2 Report

Comments and Suggestions for Authors

The authors have made proper modification to the manuscript, and it can be published.